# Assessment of Suitable Land for Surface Irrigation in Ungauged Catchments: Blue Nile Basin, Ethiopia

**Getenet Nigussie [1], Mamaru A. Moges [1,2,\*], Michael M. Moges [1]** and **Tammo S. Steenhuis [3]**

1   Faculty of Civil and Water Resources Engineering, Bahir Dar Institute of Technology, Bahir Dar Institute of Technology, Bahir Dar University, P.O. Box 26, Bahir Dar, Ethiopia
2   Blue Nile Water Institute, Bahir Dar University, P.O. Box 79, Bahir Dar, Ethiopia
3   Department of Biological and Environmental Engineering, Cornell University, Ithaca, NY 14853, USA
\*   Correspondence: mamarumoges@gmail.com

**Abstract:** Planning and decision making for new irrigation development projects requires the systematic assessment of irrigable land together with available water resources. The data required are usually not available in developing countries, and therefore a method was developed for quantifying surface water resources and potentially irrigable land in ungauged watersheds in the Upper Blue Nile Basin using Soil and Water Assessment Tool (SWAT) model and Multi-Criterion Decision Evaluation (MCDE). The method was tested using the *Lah* river basin in the *Jabitenan* district and then applied in the whole area, including ungauged areas. In MCDE, soil type, slope, land use, and river proximity were considered. Onion, Cabbage and Tomato were grown on the identified irrigable areas. The predicted monthly stream discharge agreed well with observed values, with Nash and Sutcliffe efficiencies of 0.87 during calibration and 0.68 for validation. The SWAT model calibrated parameters from the gauged catchment were used to simulate the discharge of the ungauged catchments. The potential irrigable land was determined in *Jabitenan* woreda and included the Rivers like *Birr, Tikurwuha, Gunagun, Leza Lah, Geray, Arara, Debolah, Guysa,* and *Silala*, with an area of 460 km². By evaluating gross irrigation demand of irrigable land with available flow in rivers (both observed and simulated), the actual surface irrigation potential was 47 km². The main limitation for surface irrigation in all districts was the available water and not the land suitable for irrigation. Therefore, the study suggests that in order to irrigate a greater portion of the irrigable land, water should be stored during the monsoon rain phase for use in the last part of the dry phase.

**Keywords:** SWAT; Multi Criterion Decision Evaluation; Irrigation potential; Blue Nile Basin

## 1. Introduction

Irrigation is a key driver in sustainable development and poverty reduction. Irrigation water can be obtained from a river or pumped from a well [1]. The annual groundwater potential is 40 Gm³ a⁻¹ and the total annual discharge is 122 Gm³ a⁻¹ [2,3]. There are twelve major river basins, of which nine are wet and three are dry. The Blue Nile basin in the Ethiopian highlands provide over 80% of the water that is used in Sudan and Ethiopia by way of the Nile. Despite this abundance of water, Ethiopia receives food aid for about 10% of the population [4] because the surface water supply is spatially and temporally variable, with little available at the end of the dry monsoon phase [5].

The agricultural economy of Ethiopia is largely based on rain-fed agriculture, which employs 85% of the population, and contributes approximately 50% of the gross domestic product [6]. Around 74 million hectares, or 66% of the country, is suitable for agriculture [7]. Farmers have not been able to increase food supplies faster than the growth of the population [8].

Studies have shown that using irrigation to increase agricultural production could be one of the main drivers to end poverty [5]. However, only 4~5% (3.35 million ha) of the land that could potentially be irrigated has been developed [5]. Increased utilization of water resources for irrigated agriculture could overcome existing food shortage and overcome poverty by growing food crops during the dry phase on agricultural land that is otherwise is bare.

One of the challenges in Ethiopia in effective water resources development for irrigation has been the lack of hydrological data, because only few rivers have been gauged. Watershed models have been used to extend the available data over a larger area and serve as a means of organizing and interpreting research data. The main objective is, therefore, to develop a technique that can be employed for effective irrigation development. Specifically, this study will: (1) develop, calibrate, and validate a watershed model to predict the streamflow available for irrigation in ungauged catchments; (2) identify suitable irrigable area in both gauged and ungauged catchments; (3) evaluate the potential irrigable area with surface water potential for ungauged catchments.

The annual groundwater potential in Ethiopia is 40 Gm$^3$ a$^{-1}$ and the total annual discharge is 122 Gm$^3$ a$^{-1}$ [2,3]. There are twelve major river basins, of which nine are wet and three are dry. The Blue Nile basin in the Ethiopian highlands provide over 80% of the water that is used in Sudan and Ethiopia by way of the Nile. Despite this abundance of water, Ethiopia receives food aid for about 10% of the population [4] because the surface water supply is spatially and temporally variable with little available at the end of the dry monsoon phase [5].

The study area selected is the Jabitenan district, which has a high water resource potential and limited irrigation, with no previous resource potential assessment and evaluation, and where the water resource potential assessment has not been evaluated. The watershed model selected was the Soil and Water Assessment Tool (SWAT), which is widely used in the developing world. The developed methodological framework has the potential to be applied elsewhere as a key driver for sustainable development and poverty reduction. Irrigation water can be obtained from a river or pumped from a well [1].

## 2. Materials and Methods

### 2.1. Description of the Study Area

The study took place in the *Jabitenan* district (called "Woreda" in Ethiopia, Figure 1). It is located between the latitudes of 10°24′18″ N and 10°56′01″ N and longitudes of 37°04′12″ E and 37°30′21″ E. The district covers an area of about 12,000 km$^2$ and is located in the Blue Nile basin. The area is extensively cultivated. The district is divided into 10 catchments, three of which are gauged (Lah, Birr, and Leza), and the remaining seven are ungauged (*Geray, Guysa, Gunagun, Debolah, Silala, Arara, and Tikurwuha*). The district has a unimodal pattern of rainfall with a rain phase from June to September. The average annual rainfall for 2006 to 2015 was 1023 mm a$^{-1}$. The maximum mean monthly temperature is 35 °C and occurs at the end of the dry phase in April and May; the minimum monthly temperature is 7 °C during the beginning of the rain phase in June. Elevation ranges from 1303 m at the outlet to 2697 m in the mountains in the south-west. Most of the slopes ranges are between 2–8%, with an area of 1144 km$^2$. Steep slopes between 30 and 60% cover the remaining area. The soils are volcanic in origin. The soil texture is mainly, clay with a small portion of clay loam. Haplic Alisols is the main soil type, which accounts for 50% of the district area and Eutric Cambisols has a minimum coverage of nearly 0.2%. About 70% percent of the basin is cultivated, 3% is forest, 16% consists of shrubs with grass, 10% is grassland, and 1% is urban. The rainfed crops grown are teff, maize, pepper, and barley. High value irrigated crops include pepper, tomatoes, onions, potatoes, and other vegetable crops and chat.

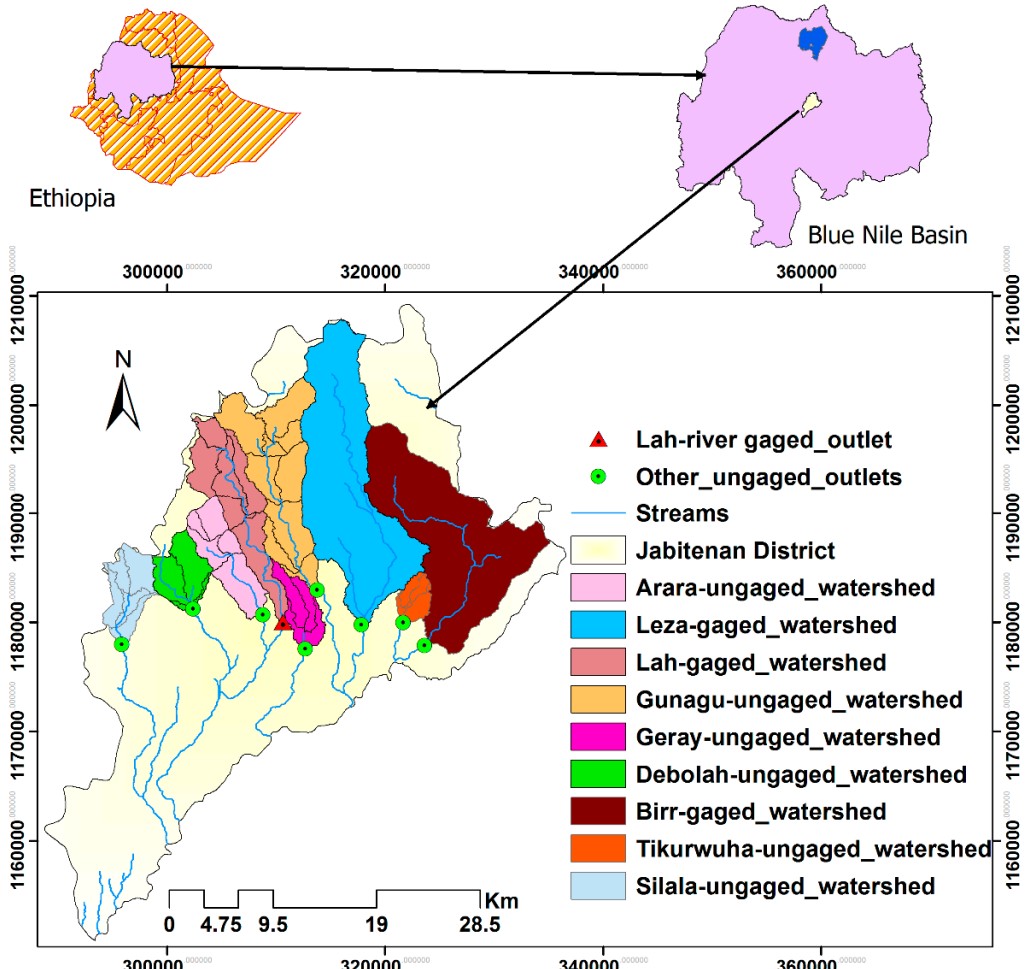

**Figure 1.** Location map of the *Jabitenan* District in Ethiopia.

## 2.2. Data Collection

### 2.2.1. Climate

Daily rainfall (1990–2014), temperature (1990–2014), wind speed (1986–2015), sunshine hours (1989–2015), and relative humidity (1993–2015) were obtained from the Ethiopian Metrological Service Agency (EMSA). The climate data were used for calibration and validation of SWAT and in CROPWAT-8 to calculate irrigation water requirements for onion, tomato, and cabbage.

### 2.2.2. Land Features

Soil and land use data were collected from the Ethiopian Ministry of Water, Irrigation and Electricity (MoWIE). A 30-m resolution Global Shuttle Radar Topography Mission (SRTM) Digital Elevation Model (DEM) (http://earthexplorer.usgs.gov/) was used to determine the percentage slope of the watershed on a pixel-by-pixel basis.

### 2.2.3. Stream Discharge

Daily stream flow data from three gauged river stations were collected from the Ministry of Water Irrigation and Electricity (MoWIE): *Birr* River near Jiga (1990–2007), *Leza* River near *Jiga* (1980–2003), and *Lah* River near *Finote Selam* town (on *Lah* River) (1985–2003). The data were checked for outliers, and missing data for the existing *Lah* River stream flow record were estimated from the records of the nearby rivers by regression using spatial statistics.

*2.3. Methods*

The general methodology includes three main parts: (i) quantifying the surface water availability during the dry phase (January to May), employing the Soil and Water Assessment Tool (SWAT) [9] model; (ii) assessing the land that is suitable for surface irrigation using the Multi-Criterion Decision Evaluation (MCDE) method in Geographical Information System, GIS [10], mainly based on topography, soils, and distance to the nearest stream [11]; and (iii) evaluating the potential surface water irrigable area.

2.3.1. SWAT Model

The SWAT model [9] is a semi-physically based model for evaluating land management practices, discharge, sediment transport, and nutrient cycling. Data inputs include precipitation, soil properties, topography, vegetation, and land management practices [9]. SWAT simulates the hydrologic cycle for each Hydrologic Response Unit (HRU) based on the following water balance Equation (1).

$$SW_t \ = \ SW_{t-1} + (R_t - Q_t - E_t - S_t - G_t) \tag{1}$$

where $SW_t$ is the final soil water content at time t, $SW_{t-1}$ is the soil water content at time $t-1$, R is the precipitation, Q is the surface runoff, E is the actual evapotranspiration, $S_t$ is the percolation and bypass exiting the soil profile bottom, and $G_t$ is the return flow [12].

Sensitivity Analysis in SWAT

Parameter sensitivity analysis was performed using the Soil Water Assessment Tool Calibration and Uncertainty Program-2, SWATCUP-2 [13] with Sequential Uncertainty Fitting-2 (SUFI2) algorithm global sensitivity methods [14]. Twenty-six hydrological parameters related to streamflow were selected based on a review of calibration parameters used in past studies [15–17], and agreeing with two recent studies in Ethiopia [18,19]. The sensitivity of the parameters was divided in four sensitivity classes from very high to low, as indicated in Table 1.

**Table 1.** Indices for sensitivity classes [20].

| Class | Index (I) | Sensitivity |
|-------|-----------|-------------|
| I | I = 1 | Very high |
| II | 0.2 < I < 1 | High |
| III | 0.05 < I < 0.2 | Medium |
| IV | 0.00 < I < 0.05 | Small to negligible |

Calibration and Validation of SWAT

The values for the most sensitive parameters were optimized with automatic calibration using the SWATCUP-2 SUFI-2 algorithm to obtain the best fit with the discharge of the Lah River. The calibration procedure was carried out on a monthly basis with half of the observed data continued until a "good" or better model performance was obtained with Coefficient of Determination, $R^2 > 0.65$, Nash Sutcliff, NS > 0.65, and Percent Bias, PBIAS > ±15 [21,22] (Table 2). This was followed by manually adjusting the parameter to obtain a physically realistic parameter set close to the optimum best fit. Once the model parameters were calibrated, validation was performed for the remaining half of the records of the Lah River discharge.

**Table 2.** General performance ratings of simulated discharge [22].

| Performance Rating | NS | PBIAS |
|--------------------|-----|-------|
| Very good | 0.75 < NS < 1.0 | PBIAS < ±10 |
| Good | 0.65 < NS < 0.75 | ±10 < PBIAS < ±15 |
| Satisfactory | 0.5 < NS < 0.65 | ±15 < PBIAS < ±25 |
| Unsatisfactory | NS < 0.5 | PBIAS > ±25 |

SWAT Model Performance

In order to evaluate the model performance, three quantitative statistical model performance measures were used. These were percent bias (PBIAS), coefficient of determination ($R^2$), and Nash and Sutcliffe efficiency (NS). The PBIAS measures the average difference between the simulated and measured values for a given quantity over the entire calibration or validation period that was calculated,

$$PBIAS = \left[ \frac{\sum\limits_{i=1}^{n}(Q_{Oi} - Q_{Si})}{\sum\limits_{i=1}^{n}(Q_{Oi})} * 100 \right] \tag{2}$$

where $Q_{Oi}$ and $Q_{Si}$ are observed and simulated stream flow values, respectively.

The coefficient of determination, $R^2$, measures how well the simulated versus observed regression line approaches an ideal match and ranges from 0 to 1 [23].

$$R^2 = \frac{\sum\limits_{i=1}^{n}(Q_{Oi} - Q_{Om})(Q_{Si} - Q_{Sm})}{\sum\limits_{i=1}^{n}(Q_{Oi} - Q_{Om})^2} \tag{3}$$

where $Q_{Om}$ is the observed average stream flow and $Q_{Sm}$ is the simulated average stream flow.

The Nash and Sutcliffe efficiency, NS [24], measures the degree of fitness of observed and simulated data, which were also used for evaluating the model performance.

$$NSE = 1 - \left[ \frac{\sum\limits_{i=1}^{n}(Q_{Oi} - Q_{Si})^2}{\sum\limits_{i=1}^{n}(Q_{Oi} - Q_m)^2} \right] \tag{4}$$

The value of NSE ranges from one to negative infinity with one being best fit between observed and simulated streamflow.

### 2.3.2. Predicting Stream Flow for Ungauged Catchments

The final calibrated parameters were used to predict discharge and water balance components of the ungauged watershed, which has similar hydrometeorological conditions. Parameters were transferred based on spatial proximity. The rationale is that catchments that are close to each other should have similar runoff patterns, as climate and catchment characteristics change gradually in space [25].

### 2.3.3. Irrigation Suitable Land Potential

To find suitable land for irrigation, the individual suitability factors, consisting of slope, soil, land use, and available irrigation water based on distance from water source, were used as inputs for the irrigation suitability model. The factors were prepared for weighted overlay with GIS ArcMap 10.1, spatial analysis tool. are discussed below. A summary of the factors with the data sources is given in Table 3.

**Table 3.** Factors for assessment of irrigation suitable land potential with their derivation and sources.

| Factor | Specific Factor | Factor Derivation | Sources |
|---|---|---|---|
| Slope factor | Slope | Digital Elevation Model(DEM) | http://earthexplorer.usgs.gov/ |
| Soil factor | Soil drainage | Soil map | [26,27] |
| | Soil depth | Soil physical characteristics | [26,27] |
| | Soil texture | Soil physical characteristics | [26,27] |
| Land use | Land use | Land use Land cover map | Ministry of Water Irrigation and Electricity MoWIE |
| Water | River proximity | DEM and River network | http://earthexplorer.usgs.gov/ and MoWIE |

Slope Factor

Using the DEM with a 30-m resolution, slopes were classified into four classes indicating the suitability for surface irrigation [26]. Slopes from 0~2% were classified as S1, from 2~4% as S2, from 4~8% as S3, and slopes above 8% were classified as S4 (Figure 2, Table 4).

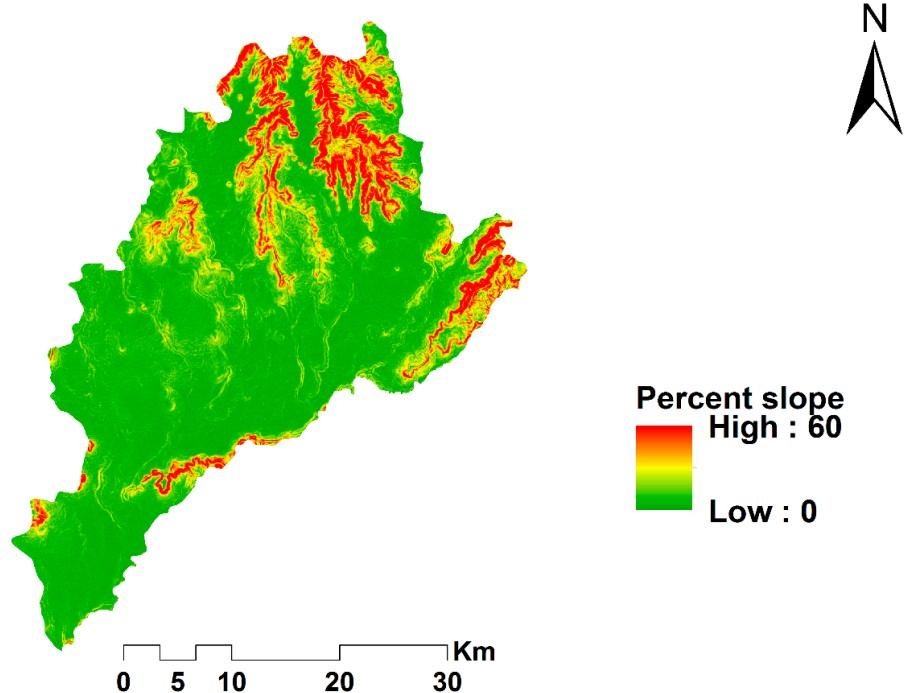

**Figure 2.** Slope map of the *Jabitenan* district in Ethiopia.

**Table 4.** Framework of land suitability classification [28].

| Suitability Order (S and N) | Suitability Class | Description |
|---|---|---|
| S | S1 (highly suitable) | Land having no significant limitation to sustained application of a given use. |
| S | S2 (moderately suitable) | Land having limitation which in aggregate are moderately severe for a sustained application of a given use |
| S | S3 (marginally suitable) | Land having limitation which in aggregate are severe for a sustained application of a given use and will reduce productivity or benefits. |
| N | N1 or S (temporarily not suitable) | Land having limitations which may be surmountable in time, but which cannot be corrected with existing knowledge at currently acceptable cost |
| N | N2 (permanently not suitable) | Land having limitations which appear as severe as to preclude any possibilities of successful sustained use of the land of a given land use. |

Soil factor

The soil map was redefined into four classes specifying its suitability for surface irrigation [27]. Soil drainage, texture, and soil depth were extracted from the soil types in the soil map for the suitability rating given in Table 4. The criteria used are provided in Table 5. According to these criteria, Dystric Leptosols and Lithic Leptosols were classified as not suitable. Eutric Cambisols, with natural fertility and good drainage, were highly suitable (Figure 3, Table 5).

**Table 5.** Soil texture suitability classification result for surface irrigation in the Jabitenan District, Ethiopia.

| Soil Code | Soil Type | Texture | Depth (cm) | Drainage | Irrigation Suitability | Area km² | Area % |
|---|---|---|---|---|---|---|---|
| FLe | EutricFluvisols | C | 125 | P | S2 | 219 | 18 |
| ALh | HaplicAlisols | C | 125 | P | S2 | 594 | 49 |
| CMe | Eutric Cambisols | C | 200 | W | S1 | 3 | 0.2 |
| LPe | Eutric Leptosols | CL | 200 | W | S2 | 21 | 1.8 |
| LPd | DystricLeptosols | C | 30 | W | N | 3 | 0.2 |
| LPq | Lithic Leptosols | C | 10 | W | N | 106 | 8.8 |
| NTh | HaplicNitisols | C | 150 | W | S1 | 249 | 21 |
| VRe | EutricVertisols | C | 125 | I | S2 | 5 | 0.4 |
| UR | Urban | | | | N | 2 | 0.2 |
| Total | | | | | | 1202 | 100 |

In texture class: C = Clay; CL = Clay Loam; in irrigation suitability class: S1 = highly suitable; S2 = moderately suitable; N = Not suitable; W = Well; I = Imperfect; P = poor.

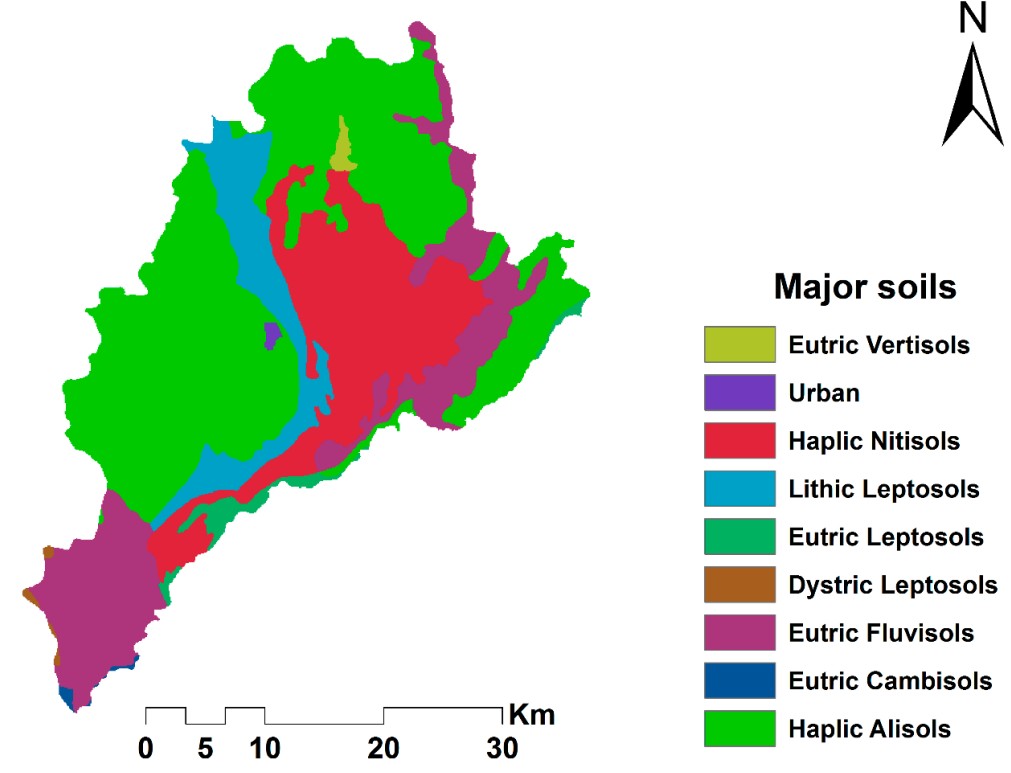

**Figure 3.** Soil suitability map of *Jabitenan* district in Ethiopia.

River Proximity Factor

River proximity (Figure 4) is another important factor that decides and prioritizes the areas to be irrigated using surface irrigation. The main perennial tributary river networks were extracted from the 30-m Digital Elevation Model (DEM) with the hydro-processing procedures in ArcGIS ArcMap 10.1.

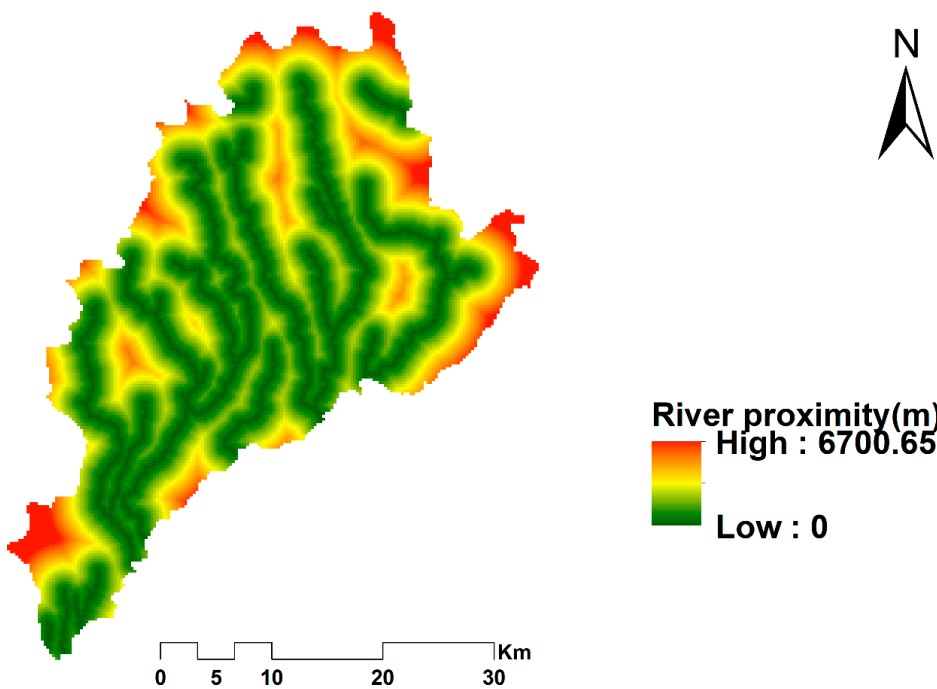

**Figure 4.** River proximity in the Jabitenan district, Ethiopia.

Land Use Suitability Factor

The suitability of the land for surface irrigation is categorized into four classes according to the Food and Agricultural Organization (FAO) framework [28], ranging from highly suitable (Class S1) to not suitable (Class S4), as shown in Table 4. Dominantly cultivated, moderately cultivated, and state farmland uses were grouped into S1, whereas wood land and urban areas were classified as S4 (Figure 5).

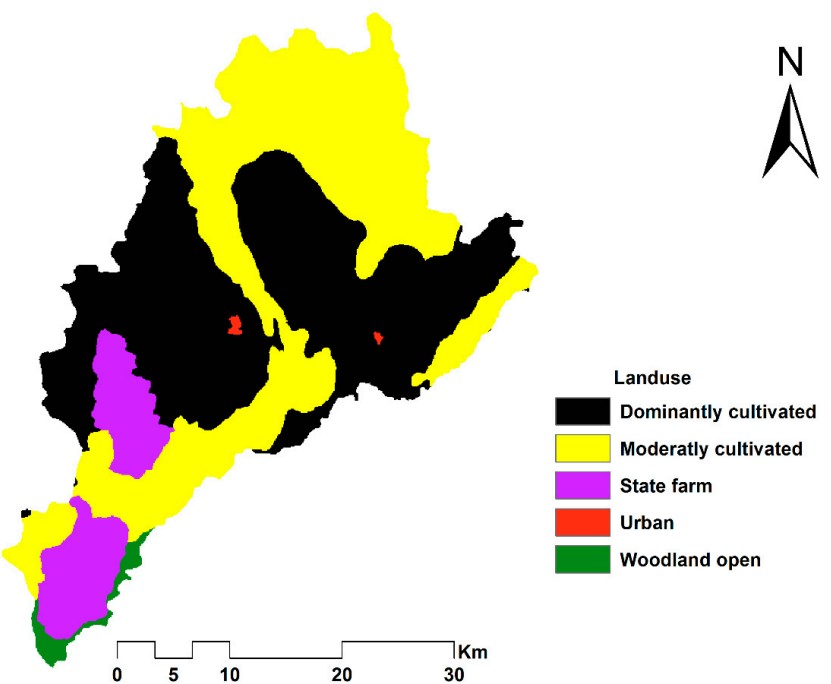

**Figure 5.** Land suitability map in the Jabitenan district, Ethiopia.

## 2.3.4. Weighted Overlay Analysis of the Factors

The weights were developed by providing a series of pair-wise comparisons of the relative importance of factors to the suitability of pixels for the activity being evaluated. The logic developed by a previous study [29] was used to produce weights under the Analytical Hierarchy Process (AHP) with a weighted linear combination. AHP was carried out by applying a weight to each parameter, which was followed by a summation of the results to yield a suitability map [30]. In pair-wise comparison, each factor was matched head-to-head (one-to-one) with each other, and a pairwise or comparison matrix was prepared to express the relative importance. A scale of importance was divided from a value of 1 to 9 (Table 6). The highest value 9 corresponds to absolute importance and the reciprocal of all scaled ratios was entered in the transpose position (1/9) and shows an absolute triviality.

**Table 6.** Pair-wise comparison matrix for weighted analysis [29].

| Intensity of Importance | Definition | Explanation |
|:---:|:---:|:---:|
| 1 | Equal importance | Two factors contribute equally to the objective. |
| 3 | Somewhat more important | Experience and judgment slightly favorable one over the other. |
| 5 | Much more important | Experience and judgment strongly favorable one over the other. |
| 7 | Very much more important | Experience and judgment very strongly favorable one over the other. Its importance is demonstrated in practice |
| 9 | Absolutely more important | The evidence favoring one over the other is of the highest possible validity. |
| 2, 4, 6, 8 | Intermediate values | When compromise is needed. |

The column factors were compared with the factors in the rows for their significance to surface irrigation, then using the scoring indicated in Table 6, a pair-wise matrix was prepared (Table 7). After the pair-wise comparison matrices were filled, the weights of the factors were computed by normalizing the respective eigenvector. The cumulative eigenvector and the weight module were used to identify the consistency ratio and develop the best-fit weights. The consistency ratio (CR) was calculated according to the methodology proposed by a previous author [31].

**Table 7.** Pair-wise comparison scale and definition [29].

| Factors | Slope | Soil Drainage | Soil Depth | Soil Texture | Land Use | River Proximity |
|:---:|:---:|:---:|:---:|:---:|:---:|:---:|
| Slope | **1** | 3 | 3 | 3 | 7 | 7 |
| Soil drainage | 1/3 | **1** | 3 | 3 | 5 | 5 |
| Soil depth | 1/3 | 1/3 | **1** | 3 | 3 | 3 |
| Soil texture | 1/3 | 1/3 | 1/3 | **1** | 3 | 3 |
| Land use | 1/7 | 1/5 | 1/3 | 1/3 | **1** | 3 |
| River proximity | 1/7 | 1/5 | 1/3 | 1/3 | 1/3 | **1** |

## 2.3.5. Irrigation Water Requirement

The reference evaporation ($Et_c$) for cabbage, onion, and tomato was calculated using CropWat8 with climatic data for the area grown using the Thiessen polygons area of influence around the weather station according to previous authors [1,32]. Irrigation water requirement (IWR) was calculated, taking into account the precipitation, P.

$$IWR \ = \ ET_c - P \tag{5}$$

Gross irrigation water requirements (GIWR) were calculated from the IWR using an irrigation efficiency ($e_a$) of 65% [33] and a water conveyance efficiency ($e_c$) of 75% [34] by multiplication with the area irrigated (A) and converting it in units of flow, as follows:

$$\text{GIWR} \;=\; \frac{\text{IWR}}{e_a \; e_c} \, \text{A} \tag{6}$$

## 3. Results and Discussion

### 3.1. SWAT Model

Sensitivity Analysis, Calibration and Validation

Following the sensitivity analysis of the 26 hydrological parameters in SWAT, 13 parameters were found to be sensitive. Of these the values of the eight most-sensitive parameters were calibrated. The rank and values for the minimum, maximum, and final optimum after calibration are shown in Table 8.

**Table 8.** Optimal/fitted parameter values and range of sensitive parameter after calibration.

| Rank | Parameter | | Minimum Value | Maximum Value | Initial | Fitted |
|------|-----------|---|---------------|---------------|---------|--------|
| 1 | SOL_Z | Depth from soil surface to bottom of layer. | 0 | 3500 | 1700 | 170 |
| 2 | CN2 | Soil Conservation Services(SCS) runoff curve number | 35 | 98 | 60 | 78 |
| 3 | CH_K2 | Effective hydraulic conductivity channel | −0.01 | 500 | 200 | 327 |
| 4 | CANMX | Maximum canopy storage | 0 | 100 | 65 | 0.8 |
| 5 | GW_DELAY | Groundwater delay | 0 | 500 | 200 | 171 |
| 6 | ALPHA_BNK | Baseflow alpha factor for bank storage. | 0 | 1 | 0.6 | 0.3 |
| 7 | CH_N2 | Manning's "n" value for the main channel. | −0.01 | 0.3 | 0.18 | 0.16 |
| 8 | REVAPMN | Revap threshold shallow aquifer wat. depth | 0 | 500 | 200 | 21 |

For the calibration period (1991–1993), the simulated monthly stream flows show a "very good" agreement with the observed monthly discharge in the Lah river, with $R^2$ = 0.92, NS = 0.87, and PBIAS = 0.2% (Figures 6 and 7). There was also a "good" agreement (Figures 8 and 9) between the measured and predicted discharge during the validation period (1994–1996), with $R^2$ = 0.77, NS = 0.68, and PBIAS = −1.4% (Table 9).

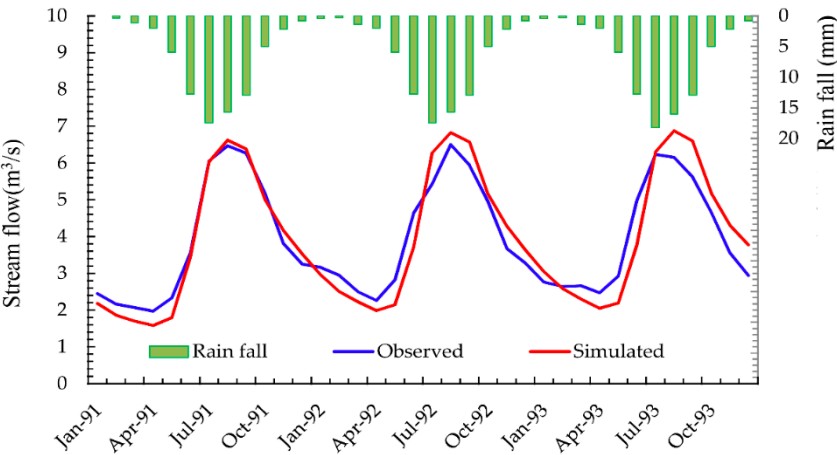

**Figure 6.** Hydrograph of simulated and observed average monthly streamflow with monthly averaged daily rainfall for calibration period for the Lah River.

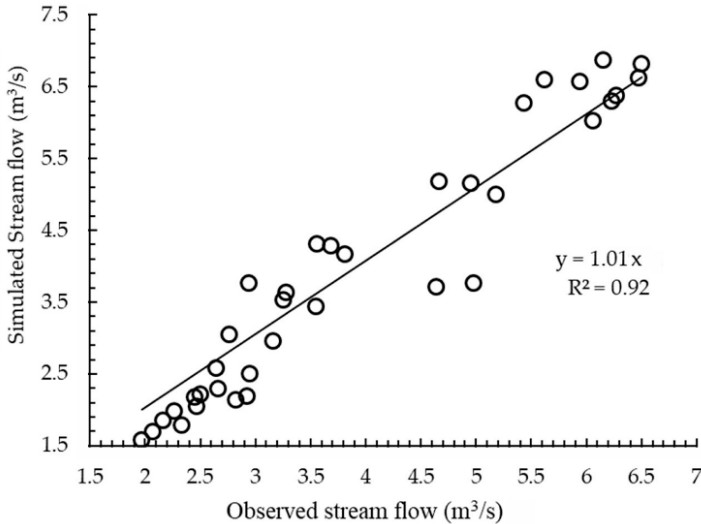

**Figure 7.** Relationship between observed and simulated discharge for calibration period for the Lah River.

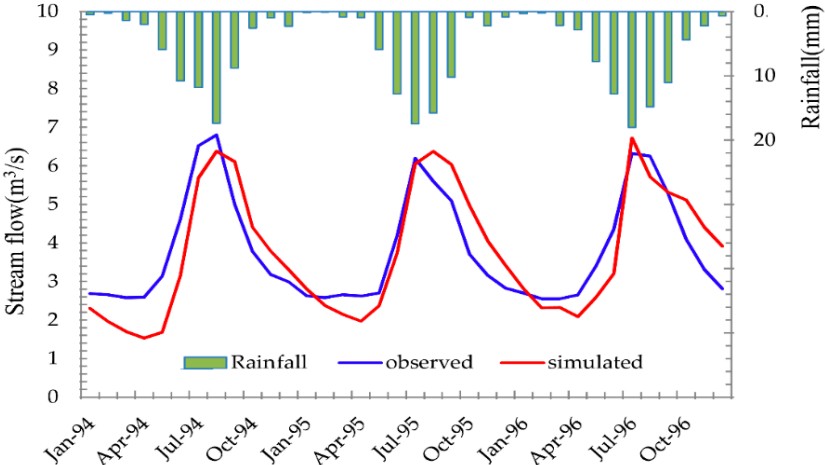

**Figure 8.** Hydrograph of simulated and observed average monthly streamflow with monthly averaged daily rainfall validation period for the Lah River.

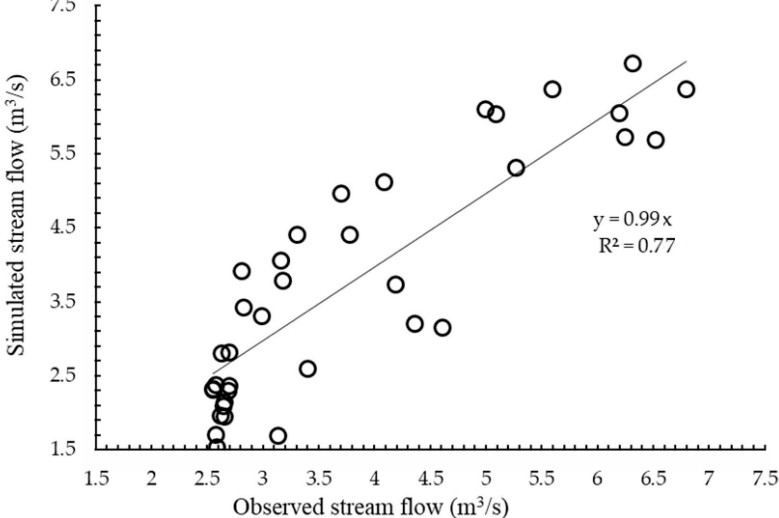

**Figure 9.** Relationship between observed and simulated discharge for the validation period for the Lah River.

**Table 9.** Model performance evaluation for calibration and validation on monthly time basis for the Lah River.

| Period | Model Performance Measures | | |
|---|---|---|---|
| | $R^2$ | NS | PBIAS (%) |
| Calibration | 0.92 | 0.87 | 0.2 |
| Validation | 0.77 | 0.68 | −1.4 |

*3.2. Ungauged Catchments Water Yield Simulation*

SWAT water yield simulation results for the period (2007–2014) for the ungauged basins showed that the annual water yield of Debolah River was the highest, followed by Silala and Geray Rivers. The annual water yields of Tikurwuha and Arara Rivers were very small (Table 8). The mean monthly predicted water yield during the dry monsoon phase (January to May), important to irrigate the selected crops, followed the same trend as the annual water yield, with the exception of the Tikurwuha river, which had relatively more low flow (Table 10).

**Table 10.** Mean monthly stream flow of ungauged river catchments in the *Jabitenan* District, Ethiopia, estimated by SWAT.

| Month | River Catchments Mean Monthly Stream Flow (m³/s) | | | | | | |
|---|---|---|---|---|---|---|---|
| | Arara | Debolah | Geray | Gunagun | Guysa | Silala | Tikurwuha |
| January | 0.33 | 0.49 | 0.5 | 0.58 | 0.27 | 0.55 | 0.23 |
| February | 0.25 | 0.41 | 0.4 | 0.46 | 0.22 | 0.44 | 0.19 |
| March | 0.2 | 0.34 | 0.32 | 0.36 | 0.19 | 0.36 | 0.15 |
| April | 0.2 | 0.3 | 0.27 | 0.35 | 0.16 | 0.33 | 0.14 |
| May | 0.23 | 0.29 | 0.29 | 0.38 | 0.2 | 0.33 | 0.16 |
| June | 0.42 | 0.96 | 1.1 | 0.63 | 0.74 | 0.85 | 0.35 |
| July | 0.99 | 1.98 | 2.39 | 1.34 | 1.42 | 1.92 | 0.68 |
| August | 1.51 | 2.47 | 2.27 | 2.23 | 1.34 | 2.61 | 0.92 |
| September | 0.89 | 1.55 | 1.79 | 1.55 | 1.28 | 1.64 | 0.63 |
| October | 0.66 | 0.97 | 1.03 | 1.14 | 0.62 | 1.11 | 0.46 |
| November | 0.5 | 0.74 | 0.76 | 0.88 | 0.39 | 0.83 | 0.35 |
| December | 0.42 | 0.61 | 0.6 | 0.74 | 0.32 | 0.69 | 0.29 |
| Annual | 6.6 | 11.1 | 11.7 | 10.6 | 7.2 | 11.65 | 4.55 |

*3.3. Irrigation Suitability Evaluation*

In order to determine the suitability of land for irrigation, we evaluated the topography, soil properties, land uses, and water resources in each area. Below we discuss each parameter.

3.3.1. Slope Suitability

Ninety-five percent of the district (covering of 1144 km²) had slopes of less than 8% and were suitable for surface irrigation systems according to FAO standard guidelines [26]; the remaining 5% of the district with slopes greater than 8% were not suitable (Figure 10a).

3.3.2. Soil Suitability

Soil properties are a major factor in the suitability of land for sustainable irrigation development. Their primary influence is on the productive capacity, but social properties also influence production and development costs. Soil texture, soil drainage, and soil depth are the most relevant physical properties of soil. Drainage suitability in is shown in Figure 10b, soil depth suitability in Figure 10c, and soil texture suitability in Figure 10d.

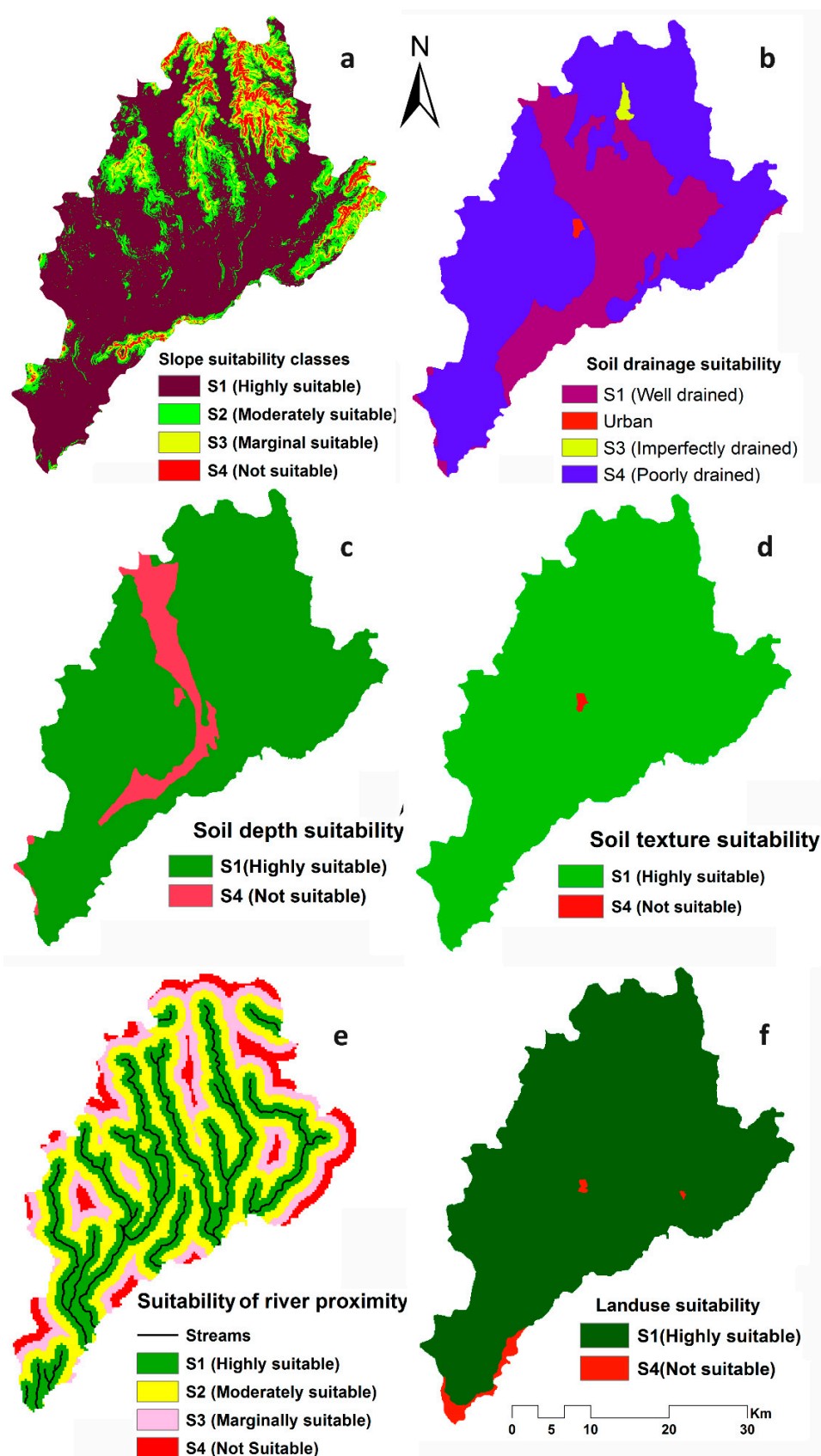

**Figure 10.** Suitability maps for the *Jabitenan* district in Ethiopia: (**a**) slope; (**b**) soil drainage; (**c**) soil depth; (**d**) soil texture; (**e**) river proximity; and (**f**) land use.

### 3.3.3. River Proximity

Suitability of the river proximity was based on dividing the proximity map of the river, with the nearest proximity assigned as highly suitable (Class S1), to not suitable (Class S4) for the farthest proximity. Figure 10e indicates the suitability class of the river proximity based on equal divisions.

### 3.3.4. Land Use Suitability

Based on the land use map of the district, it was found that 98% of the study area was suitable for surface irrigation. Only 2% of the district could not be used for surface irrigation (Figure 10f)

### 3.4. Weighting of Factors and Suitable Areas for Irrigation

The irrigation potential of the rivers was determined by weighting the slope, soil, land cover, and distance to water supply. In this study, the resulting Consistency Ratio (CR) for the pairwise comparison matrix was 0.067 (Table 11), which was acceptable for weighting the factors to evaluate the physical land capability of the *Jabitenan* district for developing an irrigation suitability map [16,31].

**Table 11.** Normalized eigenvector of the pair-wise comparison matrix.

| Factors | Weight (W) |
|---|---|
| Slope | 0.39 |
| Soil drainage | 0.25 |
| Soil depth | 0.15 |
| Soil texture | 0.11 |
| Land use | 0.06 |
| River proximity | 0.04 |
| CR | 0.067 |

Figure 11 presents the identified potential irrigable lands and Table 11 presents the identified irrigable land areas in hectares along rivers in different sites. Figure 9 clearly indicates that the highly suitable areas are found along the *Leza* and *Gunagun* Rivers. Most of the areas are classified as moderately suitable and the least as marginally suitable.

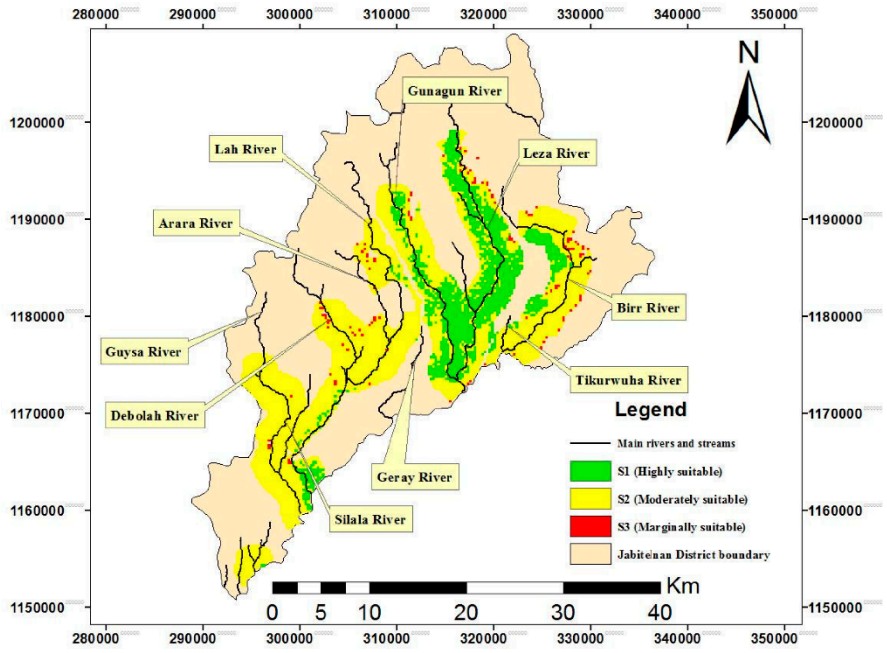

**Figure 11.** Suitable sites for surface irrigation development in the Jabitenan District, Ethiopia.

### 3.5. Irrigation Water Requirements and Irrigation Potential

The river discharge (either simulated with SWAT or observed) and the gross irrigation water requirements of onion, tomato, and cabbage for the potentially irrigable sites (Table 12) indicates that the irrigation needs of cabbage, onion, and tomato crops exceeds the minimum river discharge in all cases (Table 13). Thus, the existing water resources can irrigate only a small portion of the irrigatable land.

**Table 12.** Suitable land for surface irrigation in the *Jabitenan* District, Ethiopia.

| No | River Catchment | Command Area (km$^2$) |
|----|----|----|
| 1 | Birr and Tikurwuha | 78 |
| 2 | Gunagun and Leza | 177 |
| 3 | Lah, Geray, Arara, Debolah, Guysa, and Silala | 201 |
| | Total | 456 |

**Table 13.** Available flows of rivers and irrigation demands for cabbage, onion, and tomato in the Jabitenan District, Ethiopia.

| | River Name | | Monthly Stream Flow and Gross Irrigation Demand (m$^3$ s$^{-1}$) | | | | |
|----|----|----|----|----|----|----|----|
| | | | January | February | March | April | May |
| 1 | Birr and Tikurwuha | Available flow | 1.45 | 0.70 | 0.65 | 0.63 | 1.79 |
| | | Gross Irrigation Requirement | 1.15 | 2.90 | 4.47 | 4.77 | 2.71 |
| 2 | Gunagun and Leza | Available flow | 1.2 | 1.1 | 0.9 | 0.8 | 0.9 |
| | | Gross Irrigation Requirement | 2.60 | 6.54 | 10.08 | 10.76 | 6.12 |
| 3 | Lah, Geray, Arara, Debolah, Guysa Silala | Available flow | 4.27 | 3.75 | 3.40 | 3.30 | 3.98 |
| | | Gross Irrigation Requirement | 3.52 | 7.46 | 11.36 | 12.16 | 6.17 |

For example, the minimum available flow in the month of April for the *Birr* and *Tikurwuha* rivers in Table 13 is 0.63 m$^3$/s, whereas the water requirement of cabbage in the month of April is 4.77 m$^3$/s, giving a critical command area (that can be irrigated using the available flows in *Birr* and *Tikurwuha* rivers) of 5.2 km$^2$. Similarly, the critical command areas of other rivers were calculated in a similar way and the results are summarized in Table 14.

**Table 14.** Summary of irrigation potential of the river catchments.

| | River Catchment | Crop Type | Irrigation Potential (km$^2$) |
|----|----|----|----|
| 1 | Birr and Tikurwuha | Cabbage, onion and tomato | 5.2 |
| 2 | Gunagun and Leza | Cabbage, onion and tomato | 6.5 |
| 3 | Lah, Geray, Arara, Debolah, Guysa and Silala | Cabbage, onion and tomato | 27.2 |
| | **Total** | | **38.9** |

## 4. Conclusions

The SWAT model is a tool that can aid in overcoming data scarcity by simulating the water resource components in ungauged watersheds in Ethiopia with similar hydrometeorological conditions to those of gauged watersheds. The surface irrigation land suitability analysis indicates that only 9% of soil is not suitable for irrigation development and 5% of the land is too steep. In addition, another 4% of the land is urban, forested, or a waterbody, and cannot be used for irrigation development. Irrigation

water demand of the cabbage, onion, and tomato crops was calculated. The irrigation demand of the irrigable land for each catchment was evaluated with simulated river flow and showed that that the existing water resource potential could irrigate only a small portion of the full suitable irrigable land in the district. This implies that the irrigable land potential of the district is more limited for the available water resources than by slope, soil properties, or land use and cover. Thus, the only way to irrigate more of the irrigatable land is to prevent a greater portion of the monsoon rainfall from flowing to Sudan and to store this water so it can be used during the dry phase. The methodology developed in this manuscript can help decision makers, donors, and experts in sound irrigation development and protect downstream irrigators from losing upstream irrigation developments.

**Author Contributions:** Conceptualization, G.N., M.A.M. and M.M.M.; investigation, G.N. and M.A.M.; writing—original draft preparation, G.N.; writing—review and editing, M.A.M., M.M.M. and T.S.S.

**Funding:** This research received no much external funding.

**Acknowledgments:** However partially was supported by the Small Scale and Micro Irrigation Support (SMIS) Bahir Dar branch, where the support was funding for transportation in observed data collection. The Authors also acknowledge National Meteorological Agency of Ethiopia, Ministry of Water Irrigation and Electricity for providing hydrometerological data. We are also grateful for the *Jabitenan* district agricultural office who provided agronomy data.

**Conflicts of Interest:** The authors declare no conflict of interest.

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
