# Peer review of "Assessment of Suitable Land for Surface Irrigation in Ungauged Catchments: Blue Nile Basin, Ethiopia"

_water, doi:10.3390/w11071465_

Round 1

Reviewer 1 Report

In this manuscript, the SWAT model was implemented to ungauged basins in Upper Blue Nile River. In general, the manuscript is well-written. However, I have to suggest reject from publication for the following reasons:

1. I think similar work in this region (Blue Nile Basin, Ethiopia) has been published intensively over the past few years. The current format is a practice of the SWAT model without much in scientific depth.

Evaluation of CFSR, TMPA 3B42 and ground-based rainfall data as input for hydrological models, in data-scarce regions: The upper Blue Nile Basin, Ethiopia AW Worqlul, H Yen, AS Collick, SA Tilahun, S Langan, TS Steenhuis Catena 152, 242-251

- Evaluating hydrologic responses to soil characteristics using SWAT model in a paired-watersheds in the Upper Blue Nile Basin AW Worqlul, EK Ayana, H Yen, J Jeong, C MacAlister, R Taylor, TJ Gerik, ... Catena 163, 332-341

2. Authors should identify the study area clearly in the map (Figure 1) of Ethiopia.

3. Table 2 is over-simplified to the original version from Moriasi et al. (2007).

4. The current conclusion does not provide much insight in terms of scientific contribution(s).

Author Response

Attached is the response with the marked up mauscript. Thanks for the review.

Tammo

Reviewer 2 Report

In this paper, practical decision making is shown as a case study in Ethiopia. Although SWAT output for hydrograph is satisfied, withdrawal standard from river is highly depend on the minimum flow (or base flow). Simulation results in base flow are difference between observation and simulation. If authors would like to apply practical irrigation planning, it seems to me that parameters set should be revised. Or please try to apply uncertainty analysis in SWAT-CUP. It would be improving reliability of this paper.

Other comments are below:

Equation 5:    ea and eare not required.

In AHP, who decide weighted value? Is there any objective standard in the pair-wise comparison? If no, other combination should be examined?

Line 308, 3.1.2 >3.2.2

Line308-326  Hydrograph is showing under estimation from observed data. Minimum discharge is lower than observation. I am afraid authors are considering the model output  is applicable to irrigation planning. It means minimum discharge is so important to define the standard of withdraw. Maybe other parameter sets could be preferred to well simulated minimum river flow and annual water balance?

Line 343 3.4.1

Line 352 3.4.2

Author Response

Please find attached our responses to the comments.  The marked-up manuscript is included with the response to Reviewer 1.

Thank you for the review

Tammo

Reviewer 3 Report

I have read the manuscript and I found no novelty in it. The Authors just apply consolidated methods like SWAT and GIS techniques to assess potential areas where to develop irrigation in Ethiopia. I believe the manuscript could be interesting, provided that it is submitted to some regional journal. I do not think that publication in Water is justified.

I suggest the Authors, moreover, to control the document because English is not fluent in different points and because I feel that the manuscript has been prepared in hurry, since there are parts with different formats. 

Author Response

Our response is attached.  The marked-up manuscript is included with the response to reviewer 1

Thanks for your time in commenting on the manuscript

Tammo

Round 2

Reviewer 1 Report

The content of the manuscript has been improved significantly in the last revision. On the other hand, there are still some minor issues here and there in the manuscript. I suggest that authors should go through the manuscript very carefully and fix those minor errors before final acceptance. 

Author Response

Dear Reviewer

Thank you for your encouragement to “ go through the manuscript very carefully and fix those minor errors before final acceptance”.  We went through the manuscript fixed the minor issues and at the same time further improved the manuscript by rearranging some of the text, improve the figures and add some additional information to make the method better understandable.  The marked-up manuscript is enclosed.

Thanks so much for your time and review

Tammo and Mamaru

Reviewer 2 Report

General comment

This paper is shown application of SWAT for irrigation availability. It seems to me that the paper is suitable for this journal. Some minor revisions are recommended below.

1) Legends in all Figures are not clear. Please enhance them, larger font size are preferred.

2) Table 6, maybe I am afraid that some parameters are multiplying with initial value. For example, CN2 parameter should be between 10-90. Please check it again.

3) Figure 6 and 8, both x and y axis should begin with same numbers (Fig.6 starts from 1.5 in y and 2.0 in x, Fig.8 starts from 1 in y and 2.5 in x) and same scale.

4)  Fig.9 is hiding some text. please fit figure size.

Author Response

Dear reviewer

Please find the response to the comments in the attached file

Thanks so much

Tammo and Mamaru
